# The Relationship between Obesity and Childhood Dental Caries in the United States

**DOI:** 10.3390/ijerph192316160

**Published:** 2022-12-02

**Authors:** Érica Torres de Almeida Piovesan, Soraya Coelho Leal, Eduardo Bernabé

**Affiliations:** 1Faculty of Dentistry, Oral & Craniofacial Sciences, King’s College London, London SE5 9RS, UK; 2Department of Dentistry, Faculty of Health Science, University of Brasilia, Brasilia 70910-900, Brazil

**Keywords:** dental caries, obesity, cross-sectional studies, childhood, primary teeth, United States

## Abstract

Background: Childhood obesity and dental caries are prevalent chronic, multifactorial conditions with adverse health consequences and considerable healthcare costs. The aims of this study were: (1) to evaluate the relationship between obesity and dental caries among young children using multiple definitions for both conditions, and (2) to evaluate the role of family socioeconomic status (SES) and the child’s intake of added sugars in explaining this association. Methods: Data from 2775 2–5-year-olds children from the National Health and Nutrition Examination Survey (NHANES) 2011–2018 were analysed. Three different international standards were used to define obesity, namely the World Health Organization (WHO), Centers for Disease Control and Prevention (CDC), and the International Obesity Task Force (IOTF). Dental caries was measured during clinical examinations and summarised as counts (dt and dft scores) and prevalence (untreated caries [dt > 0] and caries experience [dft > 0]). The association of obesity with dental caries was assessed in regression models controlling for demographic factors, family SES and child’s intake of added sugars. Results: In crude models, obesity was associated with greater dt scores when using the IOTF standards (RR: 2.43, 95% CI: 1.11, 5.29) but not when using the WHO and CDC standards; obesity was associated with greater dft scores when using the WHO (1.57, 95%CI: 1.11–2.22), CDC (1.70, 95%CI: 1.17–2.46) and IOTF standards (2.43, 95%CI: 1.73–3.42); obesity was associated with lifetime caries prevalence when using the WHO (1.55, 95%CI: 1.05–2.29), CDC (1.73, 95%CI: 1.14–2.62) and IOTF standards (2.45, 95%CI: 1.61–3.71), but not with untreated caries prevalence. These associations were fully attenuated after controlling for demographic factors, family SES and child’s intake of added sugars. Conclusions: The relationship between obesity and dental caries in primary teeth varied based on the definition of obesity and dental caries used. Associations were observed when obesity was defined using the IOTF standards and dental caries was defined using lifetime indicators. Associations were fully attenuated after adjusting for well-known determinants of both conditions.

## 1. Introduction

Childhood obesity and dental caries are common chronic, multifactorial conditions that are associated with negative health effects over the life span of individuals and impose a considerable burden on national healthcare systems [1,2]. In 2020, 5.7% of children younger than 5 years old were overweight or obese, which amounted to 39 million cases worldwide [3,4]. In addition, the Global Burden of Disease study showed that the prevalence of untreated cavitated dentine caries lesions in children younger than 5 years old was 37.6% in 2019, amounting to 249 million cases globally [5]. 

Whether and how obesity and dental caries are related to each other is still a matter of debate. This is a growing area of research, with four systematic reviews published in the past five years [6,7,8,9]. While most reviews found that overweight/obesity and dental caries in older children (permanent dentition) were positively associated, the findings were less consistent in preschool children (primary dentition) [6,7,8,9]. Indeed, two reviews found that obese children showed greater caries levels in primary teeth than children of normal weight [8,9] whereas two other reviews found no association [6,7]. The mixed findings highlight the need for more studies in this specific age group, especially those that address limitations found in previous studies. One limitation consistently identified in the above reviews was the lack of adjustment for relevant confounders of the obesity-caries relationship, such as family socioeconomic status (SES) and child’s sugar intake. On one hand, both childhood obesity and dental caries are more frequent among poorer families [2,10]. On the other hand, sugar intake is an established behavioural determinant of both obesity and caries in childhood and adulthood [11,12]. The reviews also identified large heterogeneity between the primary studies. Differences in the methods used to measure obesity (e.g., standards and cut-offs to define obesity, analysis of overweight and obese groups carried out separately or jointly, and merging the underweight with the normal weight group) and dental caries (e.g., different thresholds, and analysis as count or prevalence) complicate the interpretation of findings and any comparison between studies. A detailed exploration of the impact of these methodological decisions on the estimates for the relationship between childhood obesity and caries is warranted. 

Regarding potential explanations for the obesity-caries relationship, it has been reported that obesity can lead to changes in the oral microflora and salivary properties, which in turn might predispose children to develop dental caries [13,14,15]. This explanation implies that dental caries is a consequence of obesity. However, it seems to be more accepted that the relationship between childhood obesity and dental caries is explained by common risk factors, such as a lower SES and a sugar-rich diet [9,16,17]. This second explanation has stronger implications as it would allow reducing the burden of these two common childhood health problems through policies and interventions that tackle their shared roots (e.g., sugars intake) [18]. Therefore, the aims of this study were: (1) to evaluate the relationship between obesity and dental caries in young children using multiple definitions for both conditions, and (2) to evaluate the role of family socioeconomic status (SES) and the child’s intake of added sugars in explaining this association. 

## 2. Materials and Methods

### 2.1. Study Population

This study used data from the National Health and Nutrition Examination Survey (NHANES) 2011–2018, which is a programme of studies, carried out by the Centers for Disease Control and Prevention (CDC). NHANES recruits a nationally representative survey of the non-institutionalised population in the United States through a stratified, multistage probability sampling. Data are collected annually and released for public use in 2-year cycles to protect participants’ confidentiality. The National Center for Health Statistics Research Ethics Review Board approved the NHANES survey protocol and written parental permission was obtained for participating minors. Each NHANES cycle included approximately 10,000 individuals who were interviewed at home and assessed on a variety of health factors at a mobile examination centre (MEC). Overall, there were 9756 participants in 2011–2012 (response rate: 72.6%), 10,175 in 2013–2014 (71.0%), 9971 in 2015–2016 (61.3%), and 9254 in 2017–2018 (51.9%) [19]. 

Of the 3405 children aged 24 to 71 months in NHANES 2011–2018, 3168 had complete data on body measurements and on oral health examination. Out of this total, 893 children were excluded on the basis of missing values on covariates (total energy intake = 616, child’s intake of added sugars = 593, poverty income ratio = 277, parental education = 135). Therefore, the final analytical sample for the present report was 2775 children who had complete data on all relevant variables.

### 2.2. Variables Selected

Dental caries was the study outcome, which was determined from oral health examinations by licensed dentists who were trained in the NHANES methods. Examinations were carried out at the MEC, using a portable dental chair, artificial light, and compressed air. The examination of young children included a tooth count and dental caries assessment that was registered at the cavitation-level following the Radike’s criteria [20]. Inter-examiner Kappa values for untreated caries lesions ranged from 0.93 to 1.00 [21]. The number of decayed teeth (dt score) and the sum of decayed and filled teeth (dft score) were calculated for each participant. Both indicators were also used to estimate the prevalence of untreated decay (dt > 0) and caries experience (dft > 0), respectively.

Child body measures were collected at the MEC by trained health technicians. Height was measured using a stadiometer with a fixed vertical backboard and an adjustable headpiece. Participants were weighed in kilograms using a digital weight scale wearing the standard MEC examination gown [22]. Weight and height measurements were used to classify children according to three international standards: World Health Organization (WHO), CDC and the International Obesity Task Force (IOTF). The 2007 WHO Growth reference standards for children 2 years or older were used to estimate the body mass index (BMI) for sex and age z-score, which was then categorised as underweight (lower than −2 Standard Deviations [SD]), normal weight (between −2 SD and +1 SD), overweight (higher than +1 SD, which corresponds to a BMI of 25 kg/m^2^ at 19 years), and obesity (higher than +2 SD, which corresponds to a BMI of 30 kg/m^2^ at 19 years) [23,24]. Using the CDC’s sex-specific 2000 BMI-for-age growth charts for the US child population, underweight was defined as a BMI-for-sex-and-age lower than the 5th percentile, normal weight as a BMI-for-sex-and-age equal or higher than the 5th percentile but lower than the 85th percentile, overweight as a BMI-for-sex-and-age equal or higher than the 85th percentile but lower than the 95th percentile, and obese as a BMI-for-sex-and-age higher than the 95th percentile [25]. Finally, children were classified according to the IOTF cut-offs calculated by sex and for each month of age and for the equivalent of BMI 18.5 (underweight), 25 (overweight) and 30 (obesity) at age 18 years [26].

Family SES, child’s demographic factors and intake of added sugars were also included in the analysis as potential confounders for the relationship between obesity and dental caries. Family SES was indicated by the poverty income ratio (which is estimated by dividing family income by the poverty guidelines, specific to household size, US state and year), and the household reference person’s education. Child demographic factors included were age, sex and race/ethnicity. The child’s dietary intake of all foods and beverages (including total calorie intake) was estimated from a 24 h dietary recall interview at the MEC [27]. Consumption of added sugars (in grams) was derived using the US Department of Agriculture (USDA) Food Patterns Equivalent Database (FPED) for each NHANES cycle [28]. The FPED defines added sugars as those sugars added to foods and beverages during processing or home preparation, and sugars eaten separately or added to foods at the table [29]. Intake was categorised into quartiles for analysis. 

### 2.3. Statistical Analysis

Data management and analysis were conducted in Stata version 17 (StataCorp, College Station, TX, USA). All analysis incorporated the survey design (stratification and clustering) and weights. In order to evaluate the impact of missing data on the representativeness of the study sample, children included and excluded from the analysis were compared in terms of their sociodemographic, behavioural and clinical characteristics with the Chi-square test for categorical variables and the independent samples *t*-test for numerical variables. The dt and dft scores were compared by sociodemographic and behavioural factors using simple negative binomial regression models. Similarly, the prevalence of obesity (as defined by each of three standards) was compared by sociodemographic and behavioural factors using simple binary logistic regression. 

The association of obesity with the dt and dft scores was tested using negative binomial regression models as both caries indicators were count measures with overdispersion. Therefore, rate ratios (RR) with 95% confidence intervals (95%CI) were the measure of association reported. The regression modelling started with the crude association between each definition of obesity and the dt score (labelled as Model 1). This association was controlled for the child’s demographic factors (sex, age and race/ethnicity) in Model 2 and additionally for family SES (poverty income ratio and parental education), child’s intake of added sugars and total energy intake in Model 3. The standard multivariate approach was used to adjust intake for total energy intake [30]. The same set of three sequential models was fitted for testing the association of each definition of obesity with the dft score. Finally, the same set of three models was repeated using the prevalence of untreated decay (dt > 0) and caries experience (dft > 0). Binary logistic regression was used in these analyses and odds ratios (ORs) were reported.

## 3. Results

Data from 2275 children were analysed (mean age: 3.4, SD: 1.1, range: 2 to 5 years). Statistical differences were observed between children included and excluded from the study sample due to missing values. Children in the study sample were more likely to be older and White, to have parents with higher education, and to have normal weight than those excluded. Overall, 21.9% of children had caries experience and 10.4% had untreated caries lesions. The mean dt and dft scores were 0.32 (SD: 1.46, range: 0 to 13) and 1.01 (SD: 2.89, range: 0 to 16), respectively. Older and Hispanic children, as well as those living in low SES and with greater intake of added sugars had greater dt and dft scores (Table 1). The prevalence of obesity varied based on the standards used, namely 9.5% according to the WHO standards, 11.3% according to the CDC standards, and 5.9% according to the IOTF standards. Obesity was more common among older and Hispanic children, those of low SES and with greater intake of added sugars (Table 2). 

The association of obesity with the dt score is shown in Table 3 and Figure 1. In crude models, overweight and obese children had greater dt scores than children with normal weight. In addition, estimates (RRs) were consistently higher for obese than for overweight children. However, the estimate for obesity was significant only when using the IOTF standards. Obese children had 2.43 (95%CI: 1.11, 5.29) times higher dt scores than normal-weight children. This association was weakened but continued to be significant after adjusting for child demographic factors (2.20, 95%CI: 1.07, 4.52). However, it was fully attenuated after additional adjustment for family SES and child’s intake of added sugars (1.65, 95%CI: 0.87, 3.11). Table 3 and Figure 1 also show the association of obesity with the dft score. In crude models, obese children showed greater dft scores than children with normal weight when using the WHO (1.57, 95%CI: 1.11, 2.22), CDC (1.70, 95%CI: 1.17, 2.46) and IOTF standards (2.43, 95%CI: 1.73, 3.42), respectively. However, the corresponding estimates for overweight were equal to 1 or less. The estimate for the association between obesity (according to the IOTF standards) and dft score was weakened but continued to be significant after adjustment for child demographic factors in Model 2 (2.10, 95% CI: 1.28, 3.45) and additionally for family SES and child intake of added sugars (2.02, 95% CI: 1.28, 3.17).

Similar results were obtained when modelling dental caries in terms of having untreated caries (dt > 0) and caries experience (dft > 0). Obesity was positively associated with greater odds of having caries experience, but not untreated caries, in crude logistic regression models. These associations were fully attenuated after adjustment for demographic factors and remained as such after further adjustment for family SES and child intake of added sugars (Table 4 and Figure 2).

## 4. Discussion

Little support was found for the relationship between obesity and dental caries in primary teeth among young US children. Moreover, and maybe most importantly, it was possible to observe that the findings varied depending on how obesity and dental caries were defined. For example, an association was found when obesity was defined using the IOTF standards but not using the CDC or the WHO thresholds, and when dental caries was defined using cumulative indicators, but not indicators of current untreated disease. In terms of the standards to define obesity, the highest prevalence was found with the CDC standards, which were specifically designed for US children, followed by the WHO and IOTF standards. A systematic review found that the IOTF standards yield the lowest prevalence of obesity [31]. There is evidence that the choice of the standards to define obesity can influence clinical decision to offer advice or treatment, and estimation of resources required to address the burden of obesity [31]. In that sense, an earlier review showed obesity and dental caries were associated when using the CDC standards but not associated when using the WHO standards [32]. Recent work has paved the way for harmonising estimates for the prevalence of obesity based on different standards [33] and might help to combine estimates from different studies in further systematic reviews of the association of obesity with dental caries. As for dental caries, findings were more consistent when looking at lifetime caries indicators, such as the dft score or the prevalence of caries experience, than indicators of current disease. This complicates the interpretation of cross-sectional findings as dental caries could have occurred before the development of obesity. Furthermore, whether caries is defined as a count (number of teeth affected) or prevalence (% of children affected) introduces heterogeneity in the findings and makes comparison between studies difficult. The present findings show that the use of different standards to define obesity and different summary measures to define dental caries can affect the magnitude of the association between both conditions. Researchers should be mindful of such heterogeneity when comparing findings from different studies or when pooling together estimates to carry out a meta-analysis.

Some researchers have argued that the relationship between obesity and dental caries is more commonly found in high-income countries, where both living standards and access to healthcare services (including the way in which fluorides are used) are relatively high [34,35]. The reasons behind this finding are poorly understood [34,35]. Three studies analysed NHANES data of children aged 2 to 5 years from 1999 to 2002 [36,37,38]. Two of them found that overweight or obese children had more decayed teeth than normal-weight children; however, these differences were not significant. The present finding is consistent with those from earlier studies and published reviews [6,7]. 

A second important finding of this study was that the relationship between obesity and dental caries was largely attenuated after adjusting for family SES and child intake of added sugars. These findings suggest that obesity does not increase caries risk and the presence of caries lesions does not increase obesity risk, but rather that shared determinants drive the development of both conditions [16]. Low SES (as indicated by living in families with lower poverty income ratio and with less educated parents in this study) and dietary factors (consumption of added sugars) are shared risk factors potentially linking obesity and dental caries [16,17,39]. This interpretation implies that earlier studies which did not adjust for family SES and child’s sugars intake could have overestimated the true association of obesity with dental caries in young children. 

The present findings have implications for policy and research. They point to the shared roots of childhood obesity and dental caries. Policies and interventions that address the social determinants of health (family SES) and the commercial determinants of health (marketing, distribution and accessibility of foods and beverages containing sugars) can ensure all children have a good start in life. Identifying children at risk of becoming overweight early in life may provide opportunities for family interventions to reduce the risk of both obesity and caries. It may also provide information for the development of multidisciplinary teamwork, in addition to allowing public health efforts to focus on groups that are at greater risk. Further research is needed to test whether successful interventions to tackle childhood obesity can also benefit children’s oral health. Preventing early introduction of sugars in children’s diet could be a way forward [39,40]. 

Some limitations of this study need to be discussed as well. First, no causal inferences can be drawn from the cross-sectional data used in this study. Second, participants with missing data on obesity, caries or confounders were excluded. Since included children were more likely to be older, White and normal weight and to have more educated parents than excluded children, the present findings are not fully generalisable to the whole population of pre-school US children. Third, we used the dft score instead of the dmft score (which includes missing teeth too). This was because NHANES does not collect information on the reason for teeth’s absence in young children. Therefore, our cumulative measures of caries experience are likely to be underestimated. Finally, the dietary assessment was based on parental reports of their child’s intake over a single day (i.e., 24 h dietary recall), which might not represent the habitual diet of participants. 

## 5. Conclusions

The relationship between obesity and dental caries in 2–5-year-old US children varied based on the definition of obesity and dental caries used. Associations were observed when obesity was defined using the IOTF standards and dental caries was defined using lifetime indicators (dft score or caries experience). In addition, the relationship between obesity and caries was fully attenuated after adjusting for well-known determinants of both conditions (family SES and child sugars intake).

## Figures and Tables

**Figure 1 ijerph-19-16160-f001:**
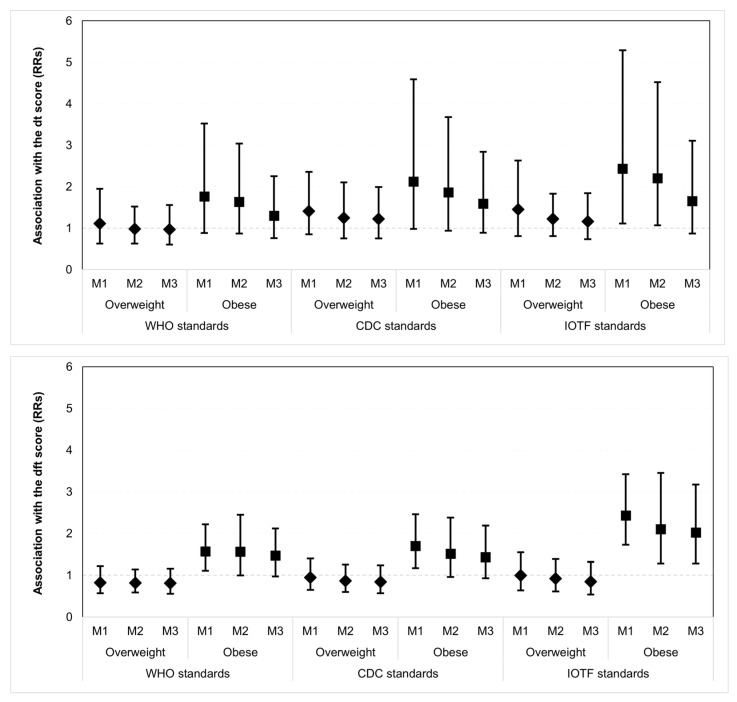
Crude and adjusted estimates (rates ratios, RRs) for the association of overweight and obesity (as defined by three international standards) with the dt and dft scores among 2–5-year-old children (*n* = 2275). M1, M2 and M3 correspond to Models 1, 2 and 3 in Table 3.

**Figure 2 ijerph-19-16160-f002:**
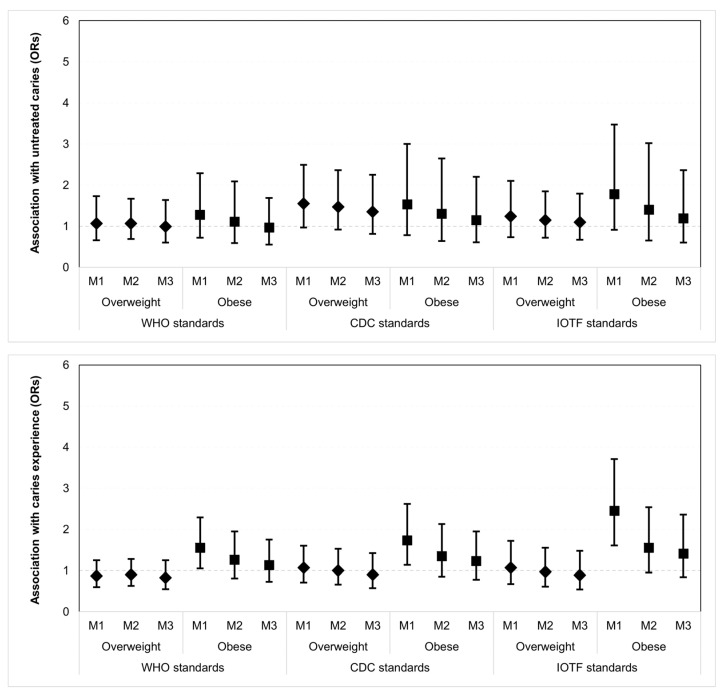
Crude and adjusted estimates (odds ratios, ORs) for the association of overweight and obesity (as defined by three international standards) with the prevalence of untreated caries and caries experience among 2–5-year-old children (*n* = 2275). M1, M2 and M3 correspond to Models 1, 2 and 3 in Table 4.

**Table 1 ijerph-19-16160-t001:** Description of the study sample and comparison of dental caries indicators by covariates.

	*n*	%	dt		dft	
			Mean	(SD)	Mean	(SD)
Child sex						
Boys	1117	49.1	0.38	(1.56)	1.10	(2.96)
Girls	1158	50.9	0.27	(1.36)	0.92	(2.82)
*p* value ^a^			0.084		0.172	
Child age						
2 years	659	24.5	0.21	(1.30)	0.27	(1.49)
3 years	533	26.1	0.29	(1.31)	0.70	(2.25)
4 years	550	24.9	0.38	(1.49)	1.11	(2.98)
5 years	533	24.6	0.41	(1.66)	1.96	(3.74)
*p* value for trend			0.025		<0.001	
Child race/ethnicity						
Non-Hispanic White	669	52.5	0.24	(0.97)	0.75	(1.89)
Non-Hispanic Black	576	13.6	0.38	(2.05)	1.06	(3.87)
Hispanic	655	23.5	0.41	(1.65)	1.54	(3.91)
Asian	193	4.0	0.52	(3.13)	1.12	(4.71)
Other	182	6.4	0.48	(2.13)	0.96	(2.87)
*p* value			0.250		0.017	
Poverty income ratio						
<1.00	817	28.1	0.59	(2.25)	1.55	(3.92)
1.00–1.99	610	25.3	0.35	(1.56)	1.01	(2.83)
2.00–2.99	317	14.9	0.22	(1.00)	1.13	(3.28)
≥3	531	31.6	0.12	(0.73)	0.46	(1.60)
*p* value for trend			<0.001		<0.001	
Parental education						
Below high school	472	15.8	0.54	(2.15)	1.92	(4.40)
High school	1246	53.5	0.41	(1.66)	1.08	(3.00)
Above high school	557	30.7	0.06	(0.50)	0.40	(1.58)
*p* value for trend			<0.001		<0.001	
Child intake of added sugars						
Q1 (lowest)	666	29.2	0.27	(1.29)	0.87	(2.60)
Q2	811	36.6	0.31	(1.44)	0.90	(2.72)
Q3	576	25.0	0.39	(1.61)	1.22	(3.26)
Q4 (highest)	222	9.3	0.36	(1.63)	1.25	(3.30)
*p* value trend			0.215		0.023	

^a^*p* values were derived from crude negative binomial regression models, using an omnibus test to compare unordered groups and a test for linear trends to compare ordered groups.

**Table 2 ijerph-19-16160-t002:** Prevalence of obesity (as defined by three international standards ^a^) by sociodemographic and behavioural factors (*n* = 2275).

	WHO Standards		CDC Standards		IOTF Standards	
	*n*	%	*n*	%	*n*	%
Child sex						
Boys	108	9.9	121	10.9	55	5.5
Girls	112	9.2	134	11.6	74	6.3
*p* value ^b^	0.665		0.723		0.531	
Child age						
2 years	58	8.8	55	8.3	19	2.7
3 years	43	6.4	52	8.1	18	3.2
4 years	52	10.1	69	13.5	36	6.5
5 years	67	12.9	79	15.2	56	11.3
*p* value for trend	0.035		0.006		<0.001	
Child race/ethnicity						
Non-Hispanic White	51	7.4	63	9.2	30	4.2
Non-Hispanic Black	58	10.7	62	11.4	32	6.9
Hispanic	90	15.2	105	17.7	55	9.9
Asian	7	3.6	9	4.7	3	1.7
O ther	14	6.5	16	8.3	9	5.2
*p* value	<0.001		<0.001		0.003	
Poverty income ratio						
<1.00	91	12.0	100	12.6	55	8.1
1.00–1.99	64	10.3	80	13.3	40	6.7
2.00–2.99	35	11.1	38	13.2	19	6.2
≥3	30	5.9	37	7.5	15	3.0
*p* value for trend	0.031		0.076		0.029	
Parental education						
Below high school	64	13.0	69	13.2	34	7.2
High school	127	10.7	151	13.4	79	7.4
Above high school	29	5.7	35	6.5	16	2.5
*p* value for trend	0.004		0.002		0.002	
Child intake of added sugars						
Q1 (lowest)	54	6.9	60	8.5	23	3.2
Q2	79	9.5	90	10.7	51	6.5
Q3	62	11.6	72	14.1	37	6.9
Q4 (highest)	25	12.2	33	14.7	18	9.0
*p* value for trend	0.097		0.036		0.044	

^a^ Obesity was defined as a BMI for sex and age z-score > +2 SD (WHO standards), a BMI-for-sex-and-age > 95th percentile (CDC standards) and a BMI ≥ those age-sex specific BMI cut-offs corresponding to BMI = 30 at age 18 years (IOTF standards). ^b^ Unordered groups were compared using the Chi-square test whereas ordered groups were compared using the Chi-square test for linear trends.

**Table 3 ijerph-19-16160-t003:** Models for the association of BMI groups (as defined by three international standards) with dt and dft scores among 2–5-year-old children (*n* = 2275).

	Mean	(SD)	Model 1	Model 2	Model 3
			RR [95%CI]	RR [95%CI]	RR [95%CI]
Untreated Caries:					
WHO standards					
Normal	0.29	(1.41)	1.00 [Reference]	1.00 [Reference]	1.00 [Reference]
Overweight	0.33	(1.27)	1.11 [0.63–1.95]	0.98 [0.63–1.52]	0.97 [0.60–1.56]
Obese	0.52	(2.10)	1.76 [0.88–3.52]	1.63 [0.87–3.04]	1.30 [0.76–2.25]
Underweight	0.97	(3.70)	3.31 [0.58–18.75]	2.58 [0.42–15.95]	1.42 [0.27–7.41]
CDC standards					
Normal	0.27	(1.29)	1.00 [Reference]	1.00 [Reference]	1.00 [Reference]
Overweight	0.38	(1.32)	1.41 [0.85–2.36]	1.25 [0.75–2.10]	1.22 [0.75–1.99]
Obese	0.56	(2.10)	2.12 [0.98–4.59]	1.86 [0.94–3.68]	1.59 [0.89–2.84]
Underweight	0.55	(2.68)	2.07 [0.85–5.06]	1.63 [0.68–3.89]	1.37 [0.67–2.81]
IOTF standards					
Normal	0.28	(1.31)	1.00 [Reference]	1.00 [Reference]	1.00 [Reference]
Overweight	0.41	(1.46)	1.45 [0.81–2.63]	1.22 [0.81–1.83]	1.16 [0.73–1.84]
Obese	0.68	(2.36)	2.43 [1.11–5.29] *	2.20 [1.07–4.52] *	1.65 [0.87–3.11]
Underweight	0.35	(1.88)	1.26 [0.67–2.37]	0.92 [0.53–1.62]	0.93 [0.52–1.68]
Caries Experience:					
WHO standards					
Normal	0.99	(2.96)	1.00 [Reference]	1.00 [Reference]	1.00 [Reference]
Overweight	0.82	(2.32)	0.83 [0.57–1.22]	0.82 [0.59–1.14]	0.81 [0.56–1.16]
Obese	1.56	(3.66)	1.57 [1.11–2.22] *	1.56 [1.00–2.45]	1.47 [0.97–2.12]
Underweight	1.08	(3.78)	1.09 [0.22–5.28]	0.69 [0.15–3.19]	0.42 [0.11–1.66]
CDC standards					
Normal	0.93	(2.82)	1.00 [Reference]	1.00 [Reference]	1.00 [Reference]
Overweight	0.88	(2.36)	0.95 [0.65–1.40]	0.87 [0.60–1.26]	0.84 [0.57–1.24]
Obese	1.58	(3.57)	1.70 [1.17–2.46] *	1.51 [0.96–2.38]	1.43 [0.93–2.19]
Underweight	1.21	(4.08)	1.30 [0.66–2.55]	1.15 [0.52–2.55]	0.83 [0.45–1.54]
IOTF standards					
Normal	0.90	(2.69)	1.00 [Reference]	1.00 [Reference]	1.00 [Reference]
Overweight	0.90	(2.37)	1.00 [0.64–1.55]	0.92 [0.61–1.39]	0.85 [0.54–1.32]
Obese	2.19	(4.12)	2.43 [1.73–3.42] *	2.10 [1.28–3.45] *	2.02 [1.28–3.17] *
Underweight	1.29	(4.15)	1.43 [0.85–2.40]	1.16 [0.67–2.02]	1.01 [0.59–1.73]

The dt and dft scores were modelled using negative binomial regression. Rate ratios (RR) were reported. Model 1 was unadjusted, Model 2 adjusted for child demographic factors (sex, age and race/ethnicity) and Model 3 additionally adjusted for family socioeconomic status (poverty income ratio and parental education) and child intake of added sugars (quintiles) and total energy intake. * *p* < 0.05.

**Table 4 ijerph-19-16160-t004:** Models for the association of BMI groups (as defined by three international standards) with the prevalence of untreated caries and caries experience among 2–5-year-old children (*n* = 2275).

	*n*	%	Model 1	Model 2	Model 3
			OR [95%CI]	OR [95%CI]	OR [95%CI]
Untreated Caries:					
WHO standards					
Normal	190	10.0	Reference	Reference	Reference
Overweight	53	10.6	1.07 [0.66–1.73]	1.07 [0.69–1.67]	0.99 [0.60–1.64]
Obese	29	12.5	1.28 [0.72–2.29]	1.11 [0.59–2.09]	0.97 [0.55–1.69]
Underweight	2	20.5	2.31 [0.38–14.0]	1.75 [0.25–12.0]	1.46 [0.27–7.94]
CDC standards					
Normal	183	9.3	Reference	Reference	Reference
Overweight	40	13.6	1.55 [0.97–2.49]	1.47 [0.92–2.36]	1.35 [0.81–2.25]
Obese	38	13.5	1.53 [0.78–3.00]	1.30 [0.64–2.65]	1.15 [0.61–2.20]
Underweight	13	10.8	1.19 [0.58–2.45]	1.17 [0.58–2.37]	1.35 [0.64–2.86]
IOTF standards					
Normal	189	9.8	Reference	Reference	Reference
Overweight	38	11.8	1.24 [0.73–2.10]	1.15 [0.72–1.85]	1.10 [0.67–1.79]
Obese	20	16.2	1.78 [0.91–3.47]	1.40 [0.65–3.02]	1.19 [0.60–2.36]
Underweight	27	10.1	1.04 [0.58–1.85]	0.95 [0.54–1.68]	1.07 [0.57–2.00]
Caries Experience:					
WHO standards					
Normal	377	21.7	Reference	Reference	Reference
Overweight	107	19.3	0.87 [0.60–1.25]	0.90 [0.63–1.28]	0.83 [0.55–1.25]
Obese	58	30.0	1.55 [1.05–2.29] *	1.26 [0.81–1.95]	1.13 [0.73–1.75]
Underweight	2	20.5	0.93 [0.15–5.67]	0.65 [0.11–3.90]	0.56 [1.22–2.61]
CDC standards					
Normal	378	20.7	Reference	Reference	Reference
Overweight	73	21.7	1.07 [0.71–1.60]	1.00 [0.66–1.53]	0.90 [0.57–1.42]
Obese	73	31.1	1.73 [1.14–2.62] *	1.35 [0.85–2.13]	1.23 [0.78–1.95]
Underweight	20	17.9	0.84 [0.43–1.63]	0.87 [0.44–1.73]	1.00 [0.56–1.80]
IOTF standards					
Normal	388	20.7	Reference	Reference	Reference
Overweight	65	22.0	1.07 [0.67–1.72]	0.97 [0.61–1.55]	0.89 [0.54–1.48]
Obese	42	39.0	2.45 [1.61–3.71] *	1.55 [0.95–2.54]	1.41 [0.84–2.36]
Underweight	49	20.3	0.98 [0.59–1.61]	0.91 [0.58–1.43]	1.03 [0.65–1.62]

Prevalence rates were modelled using binary logistic regression. Odds ratios (OR) were reported. Model 1 was unadjusted, Model 2 adjusted for child demographic factors (sex, age and race/ethnicity) and Model 3 additionally adjusted for family socioeconomic status (poverty income ratio and parental education) and child intake of added sugars (quintiles) and total energy intake. * *p* < 0.05.

## Data Availability

The NHANES data used for this report is freely available from https://www.cdc.gov/nchs/nhanes/index.htm (accessed on 23 September 2022).

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
