# Peer review of "The Relationship between Obesity and Childhood Dental Caries in the United States"

_ijerph, 2022, doi:10.3390/ijerph192316160_

Round 1

Reviewer 1 Report

The manuscript is well-written, well-organized and a pleasure to read.  It addresses the inconsistencies in much of the literature that exists that examines the relationship between obesity and Dental Caries in the primary dentition.  The reviewer is particularly impressed by the scrutiny of how different measures of obesity and Caries influences the supposed association between the two entities. 

My specific comments:

page 3 line 99  Define or provide reference for Radike's Criteria.  Readers may not be familiar with this.

Page 3 line 116  The classifying of underweight children at 5th percentile whereas the overweight children are those at 85th percentile.  The lack of symmetry ( 5 vs. 15) seems inconsistent.  Perhaps the underweight should have included those below the 15th percentile.

However most importantly in reconsidering the manuscript I found the biggest weakness is the presentation of the data in the table formats.  These tables are  rather confusing and lend little clarity to the reader trying to understand the data.  My recommendation would be to reconsider these tables and perhaps design a figure which helps the reader understand the data more easily.  Some of the tables may even be eliminated since the data is clearly described in the text.  I think the table issue will require major revision.

Author Response

Thank you for taking time to review our manuscript and for all the insightful comments. We have taken on board the recommendations made by the two reviewers and provide a point-by-point response to those comments below. We also changed the text of our manuscript accordingly to reflect such changes using the “Track Changes” function in MS Word.

REVIEWER 1

Comment: The manuscript is well-written, well-organized and a pleasure to read.  It addresses the inconsistencies in much of the literature that exists that examines the relationship between obesity and Dental Caries in the primary dentition.  The reviewer is particularly impressed by the scrutiny of how different measures of obesity and Caries influences the supposed association between the two entities. 

My specific comments:

page 3 line 99 - Define or provide reference for Radike's Criteria.  Readers may not be familiar with this.

Response: A citation was added to support the Radike’s criteria (line 100, reference 20).

Comment: Page 3 line 116 - The classifying of underweight children at 5th percentile whereas the overweight children are those at 85th percentile.  The lack of symmetry (5 vs. 15) seems inconsistent.  Perhaps the underweight should have included those below the 15th percentile.

Response: We used the CDC recommended BMI-for-age cut-offs as presented on their website: https://www.cdc.gov/nccdphp/dnpao/growthcharts/training/bmiage/page4.html This resource was cited in line 121 (reference 25) to support the cut-off points used.

Comment: However most importantly in reconsidering the manuscript I found the biggest weakness is the presentation of the data in the table formats.  These tables are rather confusing and lend little clarity to the reader trying to understand the data.  My recommendation would be to reconsider these tables and perhaps design a figure which helps the reader understand the data more easily.  Some of the tables may even be eliminated since the data is clearly described in the text.  I think the table issue will require major revision.

Response: As suggested, we have added two Figures to our manuscript reporting the estimates for the association between obesity and dental caries. We kept the original Tables 3 and 4 though for readers interested in the precise estimates rather than the visual trends.

Reviewer 2 Report

Well done for this interesting study! Kindly please address the following;

Line 53 - change "sugars" intake to "sugar intake.

Line 70 - change "share" to "shared roots".

Please indicate why you felt the need to analyze and mention your excluded sample when you discussing your demographic profile. The reasoning is not clear?

Table 2 - Please indicate the criteria's /scores of obesity according to the three standards of WHO, CDC & IOTF?? you need to mention them at some point in the paper.

In terms of the findings of your results being dependent on the various definitions..clarify on the social significance & relevance of that..it is not clear in your paper.

Author Response

Thank you for taking time to review our manuscript and for all the insightful comments. We have taken on board the recommendations made by the two reviewers and provide a point-by-point response to those comments below. We also changed the text of our manuscript accordingly to reflect such changes using the “Track Changes” function in MS Word.

REVIEWER 2

Comment: Well done for this interesting study! Kindly please address the following;

Line 53 - change "sugars" intake to "sugar intake.

Response: The text was revised as recommended (line 53).

Comment: Line 70 - change "share" to "shared roots".

Response: The text was revised as recommended (line 70).

Comment: Please indicate why you felt the need to analyze and mention your excluded sample when you discussing your demographic profile. The reasoning is not clear?

Response: We compared children included and excluded from the analysis to evaluate the impact of missing data on the representativeness of the study sample. This is now explained in the first paragraph of Statistical Analysis (lines 139-143).

Comment: Table 2 - Please indicate the criteria's /scores of obesity according to the three standards of WHO, CDC & IOTF?? you need to mention them at some point in the paper.

Response: The criteria to define obesity according to the three standards (WHO, CDC and IOTF) were presented in section 2.2. Variables selected (lines 110 to 123). For ease, we have now briefly described these criteria in a footnote to Table 2.

Comment: In terms of the findings of your results being dependent on the various definitions..clarify on the social significance & relevance of that..it is not clear in your paper.

Response: We have added a couple of sentences to the first paragraph of the Discussion to address this comment (lines 248-252). They read: “The present findings show that the use of different standards to define obesity and different summary measures to define dental caries can affect the magnitude of the association between both conditions. Researchers should be mindful of such heterogeneity when comparing findings from different studies or when pooling together estimates to carry out a meta-analysis.”